# Repeat Variants, Biomarkers, and Molecular Signatures in Parkinson’s Disease: *ATXN2*, *ATXN3*, *CACNA1A*, *PRNP*, *TBP*, *C9ORF72*, *TOMM40*, *APOE*, and *POLG*—A Swedish Perspective

**DOI:** 10.3390/ijms26189213

**Published:** 2025-09-20

**Authors:** Jose Miguel Laffita-Mesa, Martin Paucar, Per Svenningsson

**Affiliations:** 1Department of Clinical Neuroscience (CNS), Karolinska Institutet, 171 64 Stockholm, Sweden; martin.paucar-arce@regionstockholm.se; 2Department of Neurobiology, Care Sciences and Society, Karolinska Institutet, 171 64 Stockholm, Sweden

**Keywords:** Parkinson’s disease, genetic risk factors, *ATXN2*, *C9ORF72*, *APOE*, repeats expansions, Sweden

## Abstract

Parkinson’s disease (PD), the second most common neurodegenerative disorder globally, has a notably high prevalence in Sweden (136/10^5^). Although monogenic forms represent only a small subset of PD cases, several genetic factors—including nucleotide repeat expansions (NREs) in *ATXN2*, *ATXN3*, *C9ORF72*, *TBP*, *POLG*, *TOMM40*, *CACNA1A*, and *PRNP*—have been implicated in neurodegenerative conditions with parkinsonian features. However, their contribution to PD pathogenesis in the Swedish population remains understudied. We analyzed DNA from 161 Swedish PD patients and 546 controls and evaluated clinical and CSF biomarkers (tau, phospho-tau, and β-amyloid). Intermediate *ATXN2* CAG expansions were significantly associated with PD (3.40%, *p* = 0.0027), and novel promoter structural variations were identified. *C9ORF72* G4C2 expansions were also linked to PD (2.48%, *p* = 0.0018), with distinct methylation patterns in PD cases. POLG Not-10/Not-11Q alleles were positively associated (9.62%, *p* = 0.014), while *TOMM40* showed partial associations for rare genotypes (14.28%, *p* = 0.0014). Pathological expansions in *TBP* were marginally significant, while *ATXN3*, *CACNA1A*, and *PRNP* showed no associations. Two-way ANOVA identified significant interactions between *APOE E3*/*E4* and *POLG* 10/11Q genotypes, affecting age at diagnosis (*p* = 0.025) and CSF β-amyloid levels. Regression highlighted tau as a key predictor of age at diagnosis (*p* = 0.02). Longitudinally, *APOE E4* predicted cognitive decline (*p* = 0.015), and *TOMM40* haplotypes correlated with motor deficits. In conclusion, *ATXN2*, *C9ORF72*, and *POLG* emerge as key genetic risk factors for PD in the Swedish population, with *TOMM40* and *TBP* contributing partially. Altered CSF biomarker patterns support the existence of distinct molecular subtypes and warrant further investigation of novel *ATXN2* variants as potential PD modifiers.

## 1. Introduction

Parkinson’s disease (PD) is a complex, progressive, and incurable neurodegenerative condition, posing a significant health burden [1,2]. It is clinically characterized by the presence of core motor symptoms, with bradykinesia being a hallmark, accompanied by either resting tremor or muscle rigidity [3]. The condition arises due to the gradual loss of dopamine-producing neurons, especially in the substantia nigra pars compacta (SNpc). In addition to neuronal loss, the accumulation of Lewy bodies—abnormal clumps of the protein alpha-synuclein—is a key pathological feature. The underlying mechanisms thought to drive PD progression include the toxic buildup of proteins, disruption of mitochondrial function, oxidative stress, and damage to mitochondrial DNA, leading to impaired cellular energy production and abnormal cell structure. These factors together contribute to the steady decline in motor and non-motor functions seen in the disease [4,5].

As the global population ages, the number of PD cases is rapidly increasing, making it one of the fastest-growing neurological disorders in terms of disability and mortality. In 2020, an estimated 9.4 million people worldwide were affected by PD [6].

In Sweden, early studies revealed widespread use of antiparkinsonian drugs, highlighting PD as a significant public health issue [7] with reported prevalence rates between 115 and 136 per 100,000 individuals, confirming the disease’s impact [8,9]. A large twin study estimated a higher adjusted prevalence of 496 per 100,000, with men being 1.45 times more likely than women to develop PD [10]. These findings illustrate the growing burden of Parkinson’s disease in Sweden.

While most PD cases are idiopathic (iPD), it remains unclear why many patients exhibit familial clustering [11]. A growing list of studies have identified over 40 genetic loci associated with an increased risk of developing the disease [12,13,14,15]. These findings highlight the role genetic factors may play, even in cases previously considered sporadic.

Studies in Swedish PD patients have revealed strong links between specific alterations in the glucocerebrosidase (GBA1) gene and PD [16]. Additionally, although ~22% of Swedish PD patients exhibit signs of familial PD, common variants, such as *LRRK2* p.(Gly2019Ser) and *SNCA* duplication, have been found at unexpectedly low frequencies in more than 2000 PD patients [17]. This suggests that other genetic factors may play a larger role in the Swedish PD population.

Unlike single-variant alterations, Nucleotide Repeat Expansions (NREs) exhibit intergenerational instability, leading to either expansion or contraction across generations [18,19]. This instability may contribute to the phenomenon known as “missing heritability,” where genetic risk factors for diseases like Parkinson’s remain undetected. These genetic alterations were difficult to detect in the early days of whole-genome association studies, which likely explains the earlier lack of positive associations with PD. Unless candidate gene studies specifically targeted their role in PD, these alterations are only identified through a combination of short-read sequencing and bioinformatics. However, with recent advances in long-read sequencing (LRS), these alterations are becoming more readily detectable [20].

Abnormal NREs in certain genes typically lead to distinct disorders, such as ataxic conditions affecting the cerebellum or neuromuscular diseases like *C9ORF72*-ALS (C9-ALS) that target motor neurons. However, some individuals display atypical phenotypes, including parkinsonian features [21,22,23,24,25], suggesting a broader overlap in neurodegenerative pathways. This blurs the lines between conditions once thought to be entirely separate and hints at shared pathological mechanisms across these seemingly distinct disorders. A clear example of this overlap occurs with Spinocerebellar ataxia Type-2 (SCA2) repeat expansions in *ATXN2*. The pathological range in SCA2 extends from 32 to 200 CAG repeats. A cerebellar phenotype is most often observed in fully penetrant alleles with ≥35 uninterrupted CAG repeats, whereas alleles in the lower range (32–34 repeats) containing CAA interruptions consistently manifest with features indistinguishable from idiopathic Parkinson’s disease [26]. Broadening this view, individuals with NREs in *ATXN3*, *TBP*, *C9ORF72*, *PRNP*, and *POLGA* have also been reported to display a spectrum of parkinsonian phenotypes, ranging from atypical to classical forms [27,28,29,30]. This phenotypic variability raises important questions about the role of NREs in basal ganglia degeneration, a key pathological feature of PD. However, the impact of NREs varies significantly across different parkinsonian populations, adding complexity to understanding their role in neurodegeneration [19]. Yet, many Swedish PD cases remain underrepresented in such studies.

Sweden, like other Scandinavian countries, has experienced extended periods of genetic isolation, resulting in a distinct genetic landscape that may differ from other European populations in both the prevalence and expression of certain genetic diseases [31,32,33]. This isolation has likely shaped the genetic factors contributing to PD, with NREs potentially presenting a unique genetic signature that influences both the risk and manifestation of the disorder in Sweden.

As a result, PD in the Swedish population may be driven by a different combination of genetic factors, setting it apart from patterns seen elsewhere in Europe. Despite these genetic nuances, the role of NREs in Swedish PD patients remains largely unexplored. Furthermore, most studies investigating repeat expansions in PD fail to integrate clinical phenotypes and biomarker profiles of affected carriers, limiting translational insights. This study investigates the contribution of key NRE-associated genes—*ATXN2*, *ATXN3*, *CACNA1A*, *TBP*, *C9ORF72*, *PRNP*, *POLGA*, and *TOMM40*—to PD in a Swedish cohort, examining their influence on clinical phenotype and associated molecular signatures to better characterize the overt stages of the disease. By focusing on these genes, we hope to shed light on the unique genetic underpinnings of PD in Sweden, contributing to a more nuanced understanding of its pathogenesis and potential avenues for targeted interventions.

## 2. Results

Figure 1 presents the overall results of the association analysis conducted in the PD and control cohorts (demography and clinical data in Appendix A). Positive associations were observed for *POLG*, *ATXN2*, and *C9ORF72*. The positive association for CAG repeats in *TBP* was marginal. Likewise, *TOMM40* was only significant for specific alleles or genotype groups (refer to details below). Conversely, neither the CAG repeats at *loci*, *ATXN3*, and *CACNA1A* nor octapeptide repeat insertions (OPRI) in the Prion gene (*PRNP*) were found to be positively associated with PD. Two cases with one OPRI deletion, considered a polymorphism, were found, but the frequency was like controls (sizing traces in Appendix A).

### 2.1. ATXN2

One hundred sixty-one PD patients (161, 100%, 2N = 322) and 547 (100%, 2N = 1094) control DNAs were successfully genotyped (control/case ratio = 3.39). CAG repeats varied from 13–37 in the PD population, while for the controls, from 13–29 CAG repeats. The most frequent allele was 22 CAG repeats for both populations (85.11 and 89.27%). Alleles, sized with 23 CAG repeats, were the second most frequently found at 8.83% and 6.04% (Figure 2a,b). Among the PD cases, one had a CAG repeat length of 37, and another was homozygous for an intermediate allele (27/27 CAG repeats). In contrast, no individuals in the control population had CAG repeats of ≥30. Intermediate CAG repeats expansions in *ATXN2* (27–37 CAG repeats) were significantly associated with PD, occurring in 11 out of 322 alleles (3.40%) compared to 10 out of 1094 controls (0.91%). This association was statistically significant (Fisher’s exact test: *p* = 0.0027; FDR-adjusted α_c_ = 0.005 for ten tests or allelic classes) with an odds ratio (OR) of 3.83 (95% CI: 1.61–9.1; *p* = 0.0023). A post hoc power analysis confirmed the study’s ability to detect significant differences between cases (2N = 322) and controls (2N = 1094), with allele frequencies of 0.0376 and 0.00091, respectively. The study achieved 99.97% power at α = 0.05, confirming that the sample size was sufficient to detect significant differences in CAG repeat distributions between PD patients and controls.

#### 2.1.1. *ATXN2* Methylation

Methylation analysis of *ATXN2*, in both sense and antisense directions, revealed that it was generally unmethylated. Gene expression analysis was not performed due to the unavailability of sufficient material.

#### 2.1.2. Novel Variants in *ATXN2*

Four PD cases were found to carry a 9-bp duplication (g271_279delinsCGGAGCGGG) (Figure 2b). Among these, three had 22/22 CAG repeats, while one was a compound heterozygote with an intermediate allele genotype (22/27 CAG repeats). The age at diagnosis (ADx) and other clinical parameters were unremarkable and have been detailed elsewhere [34].

#### 2.1.3. Clinical Characteristics of *ATXN2* Intermediate PD Expansion Carriers

Appendix A illustrates the phenotypic manifestations associated with abnormal *ATXN2* CAG repeats within our studied population. Among these cases, 3 out of 10 were male (30%), and 86% were Swedish. All cases demonstrated favorable responses to dopamine treatment. Additionally, clinical parameters such as age at diagnosis (ADx) and Disease Duration (DD) were comparable between carriers and individuals with normal CAG repeat expansions. Specifically, for ADx, carriers had a mean of M = 70.20 (SD = 9.09) compared to M = 62.76 (SD = 10.43) for non-carriers, *t* (151) = −0.92, *p* = 0.18, ns. Likewise, for DD, carriers had a mean of M = 5.70 years (SD = 5.14) compared to M = 6.20 years (SD = 5.49) for non-carriers, *t* (151) = −0.92, *p* = 0.18, ns.

No noticeable differences were found in other clinical measures, including MDS-UPDRS (M: 47.86 (SD = 11.64)) and Hoehn and Yahr staging (M = 2.65 (SD = 0.78)), concerning genetic statuses. Notably, two subjects exhibited elevated levels of CSF tau and *p*-tau, indicating the presence of tau pathology, as illustrated in Appendix A.

Dopamine Transporter scan (DaTscan) data were available for one *ATXN2* anomalous carrier: ATX-01 (22/27). DaTscan showed reduced binding with the typical rostro-caudal gradient observed in PD. This reduced DAT binding was more severe in the striatum and less in the putamen than in the caudate, as indicated by the right and left caudate binding ratio (2.07 (z-score = −4.08 **); 1.90 (z-score − 4.24 **)); this reduction was more prominent in the left than in the right putamen (binding ratio of right and left putamen (1.69 (z-score − 4.39 **); 1.52 (z-score − 6.40 **))).

The homozygous case, ATX-05, with 27/27 CAG repeats in the *ATXN2*, consisted of patients that were 67 years old on average and presented with significant cognitive decline, reflected by a MoCA score of 8.5, and was also homozygous E3/E3 for *APOE*.

#### 2.1.4. *ATXN2* in Atypical Parkinsonism

Four patients with MSA-P (22/27 CAG repeats), CBS (22/27 CAG repeats), PSP (22/30 CAG repeats), and CBD (22/35 CAG repeats) phenotypes were identified as carriers of *ATXN2* alleles ≥27 repeats. These findings may highlight the pleiotropic role of *ATXN2* in atypical parkinsonism, extending its involvement beyond classic neurodegenerative conditions. The PSP case exhibited low but detectable methylation levels for *ATXN2*-AS (1%), as shown in the representative 2D ddPCR plot (Appendix A). The CBD case demonstrated a significant reduction in striatal DAT binding, with symmetric losses in the caudate (right: 2.60; left: 2.53; z-score for both: −2.87 *) and putamen (right: 2.27, z-score: −3.03 **; left: 2.28, z-score: −4.19 **). Our findings underscore the pleiotropic role of *ATXN2* in the parkinsonian spectrum, with intermediate CAG repeat expansions (≥27 repeats) significantly associated with PD and observed across various neurodegenerative phenotypes, including PSP, CBD, and MSA-P. These atypical cases were not included in the above association analysis for typical PD.

### 2.2. C9ORF72

All individuals in the PD cohort (100%, 161) and 518 controls (94.87%, 2n = 1036) were successfully genotyped using both the two-primer method or Triple-Primed PCR, with a control-to-case ratio of 3.21. Figure 3a displays the frequency for both cohorts, with the inset highlighting the association range for PD. The hexanucleotide repeat ranged from 2 to ≥60C9RE in PD subjects. Specifically, three intermediate alleles were identified, one containing 21 30 repeats and another with 35 repeats. Six alleles exhibited moderate expansion. Although the traces displayed the characteristic saw-tooth-like pattern, the tails were short, indicating the absence of fully expanded alleles (Figure 3b–g and Appendix A). Given the limitations of routine methods (fluorescent PCR and RP-PCR) in accurately quantifying allele size, along with the ongoing debate surrounding this issue [35], we estimated the alleles to range from intermediate (20–32 repeats) to moderately long (35–60 repeats). Regarding the controls (with a normal range of 2–30 repeats), they showed two alleles with 30 repeats, alongside two moderate long/moderate/fully expanded alleles, all in a heterozygous state for both cohorts. All ≥30C9RE alleles in the control group were identified in the DNA samples from the blood bank collection, with none found in the PD spouse control group.

Using ≥30 repeats as the threshold, abnormal C9REs were significantly associated with PD, with a prevalence of 8 out of 322 (2.48%) in the case group compared to 4 out of 1036 (0.38%) in the control group. The Fisher’s exact test yielded *p* = 0.0018, α = 0.05, and OR = 6.6 (95% CI: 2.00–22.00, *p* = 0.0022). Using different cutoffs (≥35 repeats representing moderate repeat expansions), the association with anomalous alleles linked to PD was still highly significant. For repeats ≥35, the frequency was seven out of 322 (2.17%) in the case group compared to two out of 1036 (0.19%) in the control group. The Fisher exact test yielded *p* = 0.0053 and OR = 5.73 (95% CI: 1.70–19.71), with a significant result (*p* = 0.0056) that remained significant after adjusting for multiple comparisons using FDR (α_c_ = 0.025 for two tests). Significance persisted even after excluding cases with intermediate expansions and including them in the normal allele group.

Ataxin-2 and *ApoE* genotypes for this cohort were unremarkable. However, the digenic inheritance of *TBP* and *C9ORF72* repeats were observed in one sporadic PD case, which exhibited a reduced penetrance allele with 42 CAG repeats in *TBP* and C9ORF72 repeats (≥60 repeats). Age at diagnosis was 69 year, and the course of disease was severe (MDS-UPDRS = 75, H & Y = 4) in a follow-up two years after diagnosis. This patient also suffered from moderate levels of anxiety and depression (HADS =12, MADRS = 14).

#### 2.2.1. *C9ORF72* Promoter Methylation

We investigated the blood methylation levels in a CpG island within the *C9ORF72* gene. Our analysis included 69 amyotrophic lateral sclerosis (ALS) DNA samples with full expansions of C9RE from Coriell cell repositories, five PD cases with intermediate to moderate expansions (range = 35–≥60 C9RE), 25 individuals with PD but possessing normal repeat expansions in *C9ORF72* (range = 2–18 repeats), and control DNA comprising a panel of 30 individuals with diverse ethnic backgrounds with an EDP-1 plate (repeats: HAM 006 and ECACC 94082275; repeat range = 2–15). Using ddPCR, we observed methylation patterns in PD patients carrying anomalous C9RE, as depicted by positive droplets in the 2D ddPCR plot for a representative case (C9-03, genotype 2/≥35–60RE) (Figure 3h). However, further one-way ANOVA with a Kruskal–Wallis test (K-W statistic = 93.32, *p* = 0.0001), followed by Dunn’s test and FDR control using the Benjamini–Hochberg method, revealed that methylation levels in PD are not comparable to C9ALS, as this group has significantly higher methylation levels compared to CTRLS (q = 0.0001), PD (≤60RE) (q = 0.0001), and C9PD (≥60RE) (q = 0.0021) (Figure 3i).

#### 2.2.2. Clinical Characteristics of PD Patients with RE in *C9ORF72*

Appendix A depicts the clinical phenotype for all *C9ORF72* carriers, alongside DaTscan data, available for 4 out of 8 subjects (50%). All subjects were diagnosed with PD, with a predominant representation of individuals of Caucasian descent and a male-to-female ratio of 6 to 8 (75%). The average age at diagnosis was 66.00 years (SD = 6.52), and disease duration was 6.88 years (SD = 7.97), which did not differ significantly from the C9RE non-carriers (*t* (145) = −0.92, *p* = 0.19; *t* (145) = −0.39, *p* = 0.35). They were all responsive to L-DOPA therapy, consistent with the typical late onset of PD. DaTscan results available for four patients consistently reveal dopamine transporter loss in the striatum (Appendix A). Additionally, among the subjects with ≥60C9RE, two exhibited severe cerebellar degeneration. Lastly, CSF markers, including β-amyloid, tau, and phospho-tau, were normal for all subjects with available data (50%).

Our findings highlight a significant association between anomalous hexanucleotide repeat expansions (≥30 repeats) in *C9ORF72* and PD, with increased prevalence in the PD cases compared to controls. Moderate long expansions (35–≥60 repeats) were more frequent and clinically impactful in PD, correlating with severe dopamine transporter loss in the striatum, and, in some cases, the cerebellum showed degeneration. Promoter methylation levels were significantly higher in the ALS cases with full expansions compared to the PD and controls, underscoring the distinct molecular characteristics of *C9ORF72*-related pathology.

### 2.3. TBP

We successfully genotyped all PD subjects (161 individuals, 100%, 2N = 322) and all controls (547 individuals, 100%, 2N = 1094), with the control group outnumbering the PD cohort by a ratio of 3.39. Among the PD cases, 14 individuals exhibited anomalous CAG repeat expansions: nine alleles with 41 repeats (4.35%), three with 42 repeats (genotypes, two with 35/42 and 36/42 CAG repeats, 0.93%), one with 47 CAG repeats, and one with 48 repeats (37/47 and 38/48, 0.62%). In contrast, none of the controls had ≥48 CAG repeats; the largest repeat numbers found in the controls were 41 (20 individuals, 1.83%), 43 (3, 0.27%), and 44 (1, 0.09%) CAG repeats (Figure 4a,b).

Only marginal associations were observed for ≥41–48 CAG repeats in 14 out of 322 vs. 4/1093 (4.35 vs. 0.36%) (*p* = 0.053; OR: 1.94, 95% CI: 1.00-to-3.80, *p* = 0.051). Similarly, the association for ≥42–48 CAG repeats reached nominal significance (5/322 and 4/1093 (1.5% vs. 0.36%, *p* = 0.031; OR: 4.29, 95% CI: 1.15-to-16.10, *p* = 0.030)) but did not meet the adjusted threshold (α_c_ = 0.025) after FDR correction. Likewise, the association for ≥47 CAG repeats (0.62% vs. 0%, Fisher’s exact test, *p* = 0.052; OR: 17.07, 95% CI: 0.82 to 356.59, *p* = 0.067) did not achieve statistical significance at the conventional threshold of *p* = 0.05, nor after applying FDR adjustment (α_c_ = 0.016).

To achieve sufficient power to detect a significant association with ≥42–48 CAG repeats in PD, the PD cohort size would need to increase to approximately 470–612 cases to ensure power levels of 80% and 90%, respectively. Similarly, the control cohort would need to be expanded to 944–1230 individuals. This increase in sample size, achievable for us, would ensure adequate statistical power to detect meaningful associations at the conventional *p* = 0.05 significance level, while controlling for multiple comparisons using a Bonferroni-adjusted α threshold of 0.025 for two independent tests (Appendix A).

Additionally, we identified potential digenic effects, where one allele with borderline repeat lengths (e.g., 42 CAG repeats) co-occurred with a *C9ORF72* expansion. The frequency of this co-occurrence was 0.31% in the PD patients, compared to 0% in the controls (*TBP* × *C9ORF72*; Fisher’s exact test, *p* = 0.24), as previously discussed. Six additional *C9ORF72* repeats (7–10 C9REs) were observed to be co-inherited with *TBP* expansions of ≥42–48 CAG repeats, though they did not reach the intermediate cutoff of 35 repeat units seen in the case with C9 ≥ 60 and ≥42 CAG repeats in *TBP*. Notably, five out of these seven subjects carried the *C9ORF72* ‘Risk’ haplotype for the rs3849942 T-allele, whereas none of the repeats with the longest TBP expansions (CAG = 47 or 48) did.

### 2.4. Clinical Features

All subjects were diagnosed with PD and predominantly Caucasian, with a male-to-female ratio of 9:5 (64%). The mean age at diagnosis was 62.78 years (SD = 11.54), and disease duration averaged 5.75 years (SD = 5.4), comparable to non-carriers. All responded positively to DOPA therapy. MoCA scores (M= 23.1, SD = 3.64) showed no significant differences. DAT scans consistently revealed striatal DAT loss. CSF markers (β-amyloid, tau, and *p*-tau) were available for only three subjects (Appendix A).

Our findings reveal nominal associations between anomalous *TBP* CAG repeat expansions (≥42CAG repeats) and PD, though statistical significance was not maintained after correction for multiple comparisons. Larger cohort sizes are needed to achieve sufficient power for robust detection of associations. Co-inheritance of borderline *TBP* repeats with full *C9ORF72* expansions was observed in a subset of cases, which may suggest potential digenic interactions. Clinical characteristics, including ADx, DD, and treatment response, were like those of non-carriers, though DAT scans revealed consistent striatal dopamine transporter loss.

### 2.5. POLG

All PD individuals and controls had their DNA samples successfully genotyped. Positive association of the polyQ variant in *POLG* was observed with the Not-10/Not-11Q group, showing a frequency of 9.62% (31 out of 322) vs. 5.57% (61 out of 1094) in the controls (Fisher’s exact test *p* = 0.014 significant at *p* = 0.05; α_c_ = 0.025, OR = 1.80, 95% CI 1.15-to-2.83, *p* = 0.01) (Figure 5a and Appendix A).

Clinical variables such as ADx and DD were not different between Not-10/Not-11Q compared to 10/11Q_10/11Q genotypes. No differences were found in the MDS-UPDRS, H & Y, and MoCA scores and the biochemical polypeptides (β-amyloid1-42, *p*-tau, and tau) in the CSF, as determined at disease diagnosis.

Thus, the *POLG* Not-10/Not-11Q variant was significantly associated with PD, but no differences were observed in clinical variables or CSF biomarkers between genotypes.

### 2.6. TOMM40

The *TOMM40* gene, located on chromosome 19q13.2, contains the *rs10524523* locus, known as the *TOMM40* poly-T repeat. Adjacent to *TOMM40* is the *APOE* locus, crucial for cholesterol metabolism, particularly in the central nervous system. Roses et al. [36] classified rs10524523 alleles at *TOMM40* based on repeat sequence length, denoted as short (S), long (L), and very long (VL).

All 161 cases (100%) and only 531 controls (97.07%) were successfully genotyped, resulting in a control-to-case ratio of 3.29. The ApoE-E4 haplotype was strongly associated with the presence of long alleles (Fisher’s exact test, *p* = 0.00001), confirming the observations of [36]. However, no significant association was found between the *ApoE* haplotypes with PD. Likewise, no association was found for TOMM40 haplotypes (short, long, and very long) and PD (Appendix A).

Genotypes identified in the population were L-L, L-VL, S-L, S-S, S-VL, and VL-VL, with S-VL being the most frequent in both groups (PD = 30.63%, CTRL = 32.95%) (Figure 6a). The genotype Long-poly-T/Very-Long-poly-T (L-VL) showed a strong association with PD, being significantly more prevalent in the patients (14.28%, 23/161) than in the controls (6.03%, 32/531) (Fisher’s exact test, *p* = 0.0014). This association remained significant after FDR adjustment (α_c_= 0.008, corrected for six tests), with an odds ratio (OR) of 2.60 (95% CI:1.50–4.60, *p* = 0.0010) (Figure 6a).

While PD genotypes at the *TOMM40* locus were in Hardy–Weinberg Equilibrium (PD: Chi2 = 1.26, *p* = 0.74), the control population showed significant deviation (Chi2 = 36.36, *p* = 0.0001), driven primarily by the genotypes L-L (Chi2 = 19.13, *p* = 0.0001), L-VL (Chi2 = 11.29, *p* = 0.0001), and VL-VL (Chi2 = 3.75 marginally significant, *p* = 0.05), while others (S-L, S-S, and S-VL) were in HWE (Appendix A). These results suggest potential population-specific effects or biases.

### 2.7. Clinical and Biochemical Features

ADx, DD, UPDRS, MoCA, and PD scores did not differ across genotypes (Appendix A). CSF analysis revealed that β-amyloid levels were significantly lower in the L_L genotype group compared to S_L and S_VL groups, with values for L_L (N = 2, M = 355, SD = 116.7 ng/L), S_S (N = 12, M = 1021, SD = 328.5 ng/L), and S_VL (N = 20, M = 1055, SD = 278.6 ng/L). A one-way ANOVA analysis indicated a significant effect of genotype on β-amyloid levels (F = 3.42, *p* = 0.009), with post hoc Tukey’s test revealing significant differences between L_L vs. S_S (adjusted *p* = 0.053) and L_L vs. S_VL (adjusted *p* = 0.03) (Figure 6b). Genotypes with long alleles (L-L, L-VL, and VL-VL) showed no differences for CSF β-amyloid levels compared to those with at least one short allele (S-S, S-L, and S-VL), designated as intermediate (t (62) 1.58, *p* = 0.12) (Appendix A). To create the two groups, we combined the poly-T expansion lengths of alleles from the S-S, S-L, and S-VL genotypes for the Intermediates, and those from the L-L, L-VL, and VL-VL genotypes into Long. The following values were obtained for the Intermediate genotypes: N = 226, poly-T length M = 20.3, and SD = 8.8. The following values were obtained for the Long genotypes: N = 96, poly-T length M = 32.20, and SD = 4.01. No significant findings were observed for tau and phospho-tau levels.

In conclusion, while *TOMM40* alleles alone are not directly associated with PD risk, specific genotypes involving long and very long alleles contribute significantly to risk in the Swedish population. This association may reflect selective assortment of these alleles within the population, potentially influenced by non-random mating or demographic factors. Moreover, the findings highlight the potential role of *TOMM40* genotypes in influencing both PD risk and β-amyloid levels in CSF.

### 2.8. Cross-Sectional Insights into Gene Interactions and Their Impact on PD

#### 2.8.1. *APOE* × *POLG* and *TOMM40* × *POLG*

We conducted two independent two-way ANOVA analyses to examine the interactions between specific genetic variants and their effects on key biomarkers. The first analysis focused on the *APOE* haplotypes (E3 vs. E4) and *POLG* haplotypes (10/11Q_10/11Q vs. Non-10/11Q_10/11Q), while the other analysis assessed the *TOMM40* haplotypes (Intermediate vs. Long) in conjunction with the *POLG* haplotypes (see description in the Section 3). These analyses allowed us to evaluate both the main effects of the *POLG* and *TOMM40* haplotypes, as well as their interactions, on important disease-related biomarkers.

#### 2.8.2. *APOE* × *POLG* Analysis

This analysis revealed significant positive associations for ADx, beta-amyloid, and tau levels in CSF for the PD subjects. Although trends were observed for MoCA, this did not reach statistical significance (F (1, 128) =3.42; *p* = 0.06). For ADx, the two-way ANOVA revealed a significant interaction effect between the *APOE* haplotypes (E3, E4) and *POLG* genotypes (10/11Q_10/11Q, Non-10/Non-11Q_10/11Q) on the age at diagnosis, accounting for 2.60% of the total variation (F (1, 146) = 4.02; *p* = 0.046)). This suggests that the combination of these genetic factors significantly influences the age at which individuals are diagnosed. However, the main effects of the *APOE* haplotypes (*p* = 0.85) and *POLG* genotypes (*p* = 0.30) were not significant, contributing only 0.02% and 0.67% of the total variation, respectively.

Post hoc analysis using Tukey’s multiple comparison test further examined these interactions, suggesting that the observed variation was primarily driven by a cross-interaction between the *APOE* and *POLG* groups. Specifically, a significant difference was identified between the E3:10/11Q_10/11Q group (N = 78) and the E3: Non-10/Non-11Q_10/11Q group (N = 19), with a mean difference of −7.32 years (95% CI: −14.02-to-0.62, *p* = 0.025). Individuals with the E3/11Q_10/11Q genotype had a significantly earlier age at diagnosis (60.03 ± 10.17 years) compared to those with the E3: Non-10/Non-11Q_10/11Q genotype (67.95 ± 8.70 years) (see Figure 7a,d and Appendix A for full comparisons of all parameters).

The age at diagnosis showed no significant differences between the *APOE* E3 and E4 haplotypes or between the intermediate and long *POLG* genotypes and the TOMM40 haplotypes (*p* > 0.05, Appendix A), suggesting that the interaction effect is primarily driven by the specific comparison between these subgroups.

No interaction effects were observed for CSF beta-amyloid or phospho-tau levels. However, for beta-amyloid, a significant main effect was found with E3:10/11Q_10/11Q carriers (N = 33, M = 1064 ng/L, SD = 286) showing higher levels than E4:10/11Q_10/11Q carriers (N = 20, M = 748 ng/L, SD = 328; adjusted *p* = 0.0024). For phospho-tau, a trend-level difference (adjusted *p* = 0.06) suggested slightly higher levels in the E4:10/11Q_10/11Q carriers compared to their E3 counterparts, warranting further investigation (Figure 7b,c,e,f).

#### 2.8.3. *TOMM40* × *POLG* Analysis

The *TOMM40 × POLG* two-way ANOVA revealed no significant interaction or main effects for any biomarkers, including age at diagnosis, beta-amyloid, or phospho-tau levels in CSF. Post hoc analyses confirmed no subgroup differences (all *p* > 0.05, Appendix A), suggesting that *TOMM40* haplotypes alone or with *POLG* genotypes do not significantly influence these biomarkers in this cohort.

Thus, *APOE* and *POLG* interactions significantly influenced age at diagnosis and beta-amyloid levels, while *TOMM40* and *POLG* showed no impact on clinical outcomes or biomarkers in this cohort.

### 2.9. Regression Analysis and Correlation Analysis

We conducted a multiple linear regression analysis to assess the effects of genetic and clinical variables on the dependent variable, Age at Diagnosis (ADx). The model (ADx ~ Intercept + REPEATS + *APOE E4* + *TOMM40* + *POLG* + MALE + DD + Heredity + UPDRS + H & Y + MADRS + MoCA + β-Amyloid + *p*-Tau + Total Tau) was statistically significant (F (14, 32) = 2.07, *p* = 0.044)), with R^2^ = 0.4751 and adjusted R^2^ = 0.245.

Among the predictors, *TAU* levels were significant contributors (F (1, 32) = 5.98, *p* = 0.02). REPEATS and Disease Duration (DD) showed trends toward significance (*p* = 0.064 and *p* = 0.063, respectively). Other predictors, including *APOE* E4, *TOMM40*, *POLG*, and clinical parameters (e.g., MoCA, MADRS, and UPDRS), were not significant (*p* > 0.05) (Appendix A). Residual variance (SS = 2563) suggests additional unaccounted factors influencing ADx. These findings highlight the importance of tau as a key factor in the variation of ADx (Appendix A).

### 2.10. Correlation Analysis

To further explore the relationships among variables, we performed a Pearson correlation analysis. Tau showed a significant positive correlation with ADx (Pearson R = 0.324, *p* = 0.009). MoCA exhibited a significant negative correlation with ADx (R = −0.238, *p* = 0.045). *p*-tau had a near-significant positive correlation with ADx (R = 0.237, *p* = 0.057). Significant correlations were also observed between phospho-tau and tau (R = 0.752, *p* = 0.001) and between MoCA and MADRS (R = −0.296, *p* = 0.029). Variables such as REPEATS, *APOE* E4, and *TOMM40* showed weak or non-significant correlations with ADx (*p* > 0.05) (Figure 7g; QC and parameters for the correlations and linear regression are detailed in Appendix A).

Overall, our analysis explored the interplay between genetic, clinical, and biomarker variables to understand their influence on PD progression and ADx. Tau levels stood out as a significant predictor of age at diagnosis, while other factors, such as repeats and DD, showed trends toward significance. These findings underscore the importance of tau and its associated biomarkers in disease progression, while also highlighting the need for further research to uncover additional contributing factors.

### 2.11. Long-Term Effects of POLG, APOE, and TOMM40 on PD

We conducted a longitudinal study examining the association between genetic haplotypes in three genes—*APOE*, *TOMM40*, and *POLG*—and key clinical parameters: MoCA and UPDRS. These genes were selected due to their ample sample sizes and the availability of comprehensive data at two time points.

The analysis revealed significant cognitive decline (MoCA) in the *APOE* gene, specifically within the E4 haplotype group (t (41) = 2.63, *p* = 0.015, q = 0.0162 adjusted by the Benjamini–Yekutieli method (BKY); see Figure 8a). This group showed a decline of four points per year in cognitive function (M = 24 SD = 3.83 at T1 to M = 20 SD = 5.63 at T2; Figure 8i and Appendix A). However, no significant changes were observed for the E3 haplotype or in the genes *TOMM40* and *POLG* when analyzed independently (Figure 8b,c and Appendix A).

In terms of interactions, a significant decline in cognitive function was noted only in the *APOE–POLG* combination for the E4-Q1 haplotype (Figure 8k), which paralleled the decline observed in the APOE E4 group alone (t = 2.84, df = 18, *p* = 0.01, q = 0.03 after FDR adjustment).

For the UPDRS scores, one-way ANOVA with Geisser correction revealed significant differences among the haplotypes. For *APOE*, significant motor deficits were associated with the E4 haplotype, showing an increase in scores from T1 (39.89 ± 16.94) to T2 (52.33 ± 23.22), with t (18) = 3.49, *p* = 0.002, and an FDR-adjusted q-value of 0.003. The E3 haplotype showed no discovery (q > 0.05), with a non-significant finding (*p* = 0.096). Both haplotypes of *TOMM40* exhibited marked motor deficits, with the long haplotype showing an increase from INT-T1 to INT-T2 by 11 UPDRS points (±3.42), t (36) = 2.109, and *p* = 0.042 and the intermediate haplotype increasing by 4.3 points (±2.03), t (18) = 3.214, and *p* = 0.0051. *POLG* showed scores rising from T1 (38.14 ± 17.68) to T2 (45.14 ± 20.23), t (41) = 3.418, and *p* = 0.0014, with adjusted FDR (Figure 8d–f,k and Appendix A).

Significant results were also observed in interactions between *TOMM40-POLG* and specific items of UPDRS. These differences varied subtly among the groups across UPDRS Items I, II, and III (Figure 8i and Appendix A).

Overall, our analysis identified the *APOE* E4 haplotype as a reliable marker for cognitive decline. We also found that interactions between *APOE* and *POLG* genes affect cognition in a similar way. However, the study showed that motor deficits are more commonly associated with *TOMM40* haplotypes.

## 3. Discussion

Briefly, through a comprehensive analysis, we identified significant associations between PD and intermediate repeat expansions in *POLG*, *ATXN2*, and *C9ORF72*, as well as novel structural variations in the *ATXN2* promoter region. Additionally, distinct methylation patterns emerged for *C9ORF72* expansions, reinforcing the complex genetic landscape underlying PD in this population. Our findings also highlight the contribution of *POLG* alleles to PD risk, while *TOMM40* and *TBP* showed partial associations. Conversely, we found no significant links between PD and repeat expansions in *ATXN3*, *CACNA1A*, or *PRNP*. We also found significant contributions of the genes *APOE*, *POLG*, and *TOMM40* in key phenotype aspects of PD, specifically, age at diagnosis, cognitive function (MoCA), clinical scales (UPDRS), and biochemical endophenotypes (β-amyloid and tau).

Previously, a collection of thirty studies (search strategy in Appendix A) surfaced in the genetic of Parkinson disease in Sweden, with alterations in the following genes: *GBA*, *SNCA*, *POLG*, *PLPP4*, *HFE*, *LRRK2*, *S100B*, *PARK16*, *SLC45A3*, *NUCKS1*, *RAB7L1*, *SLC41A1*, *PM20D1*, *NFE2L2*, *GRIN2A*, *HLA-DRA*, *MAPT*, *GPNMB*, *CCDC62/HIP1R*, *SYT11*, *GAK*, *STX1B*, *MCCC1/LAMP3*, *ACMSD*, *FGF20*, *COMT*, *C9ORF72*, *POLG*, *GSTM1*, *NAT2*, *GSTP1*, *PSEN2*, *CYP2E1*, *UCH-L1*, *ERβ*, *CALCA*, *BDNF*, and *PON1*. Except for two studies that independently analyzed *POLG* and *C9ORF72* (see below), most of the existing research does not explore NREs/STRs in PD, nor do they assess clinical correlates. This oversight adds both novelty and translational value to our study, potentially bridging the gap between genetic findings and clinical applications.

Among the studied genes, *GBA1* (variants: E326K, N370S, and L444P), *LRRK2* (G2019S), *NFE2L2*, *GRIN2A*, *POLG*, and *UCH-L1* (S18Y) have shown a positive association or protective function against PD. Conversely, *GBA* (T369M), *HLA-DRA* (rs3129882), *C9ORF72*, and *COMT* (Val158Met) have demonstrated negative associations. Meanwhile, the roles of *SNCA*, *PARK16*, *BMP6*, and *CALCA* remain inconclusive.

Alterations in *GBA1* (E326K, N370S, and L444P) are the most frequently observed and are strongly associated with PD, particularly the L444P variant, which exhibits a significant effect size and confers an eight-fold increased risk of developing the disease compared to controls [16]. In contrast, another study reported a low frequency of the most common mutations in a well-powered (99.9%) PD sample, with *LRRK2* p.(Gly2019Ser) present in only 0.11% of cases and *SNCA* duplications in 0.045% [17]. The scarcity of monogenic causative genes in PD (*SNCA*, *PRKN*, *PINK1*, and *DJ-1*) is striking and suggests that much of the genetic architecture may be attributed to susceptibility genes, unless novel monogenic variants are uncovered in the future. In contrast to the low frequency of PD-associated genes, the predominance of L444P mutation carriers in northern Sweden [16] highlights a distinct and region-specific genetic landscape for PD. This regional disparity may be due to founder effects, which is the case for Gaucher’s disease in northern Sweden. Our study contributes to the list of PD susceptibility genes in Sweden, including *POLG* (9.6%), *ATXN2* (3.1%), and *C9ORF72* (2.5%). Although these genes are not monogenic causes of PD, they can be considered risk factors and modifiers of the disease phenotype. This adds translational value by enhancing cohort stratification for clinical trials and improving diagnostic approaches.

Anvret et al. [37] reported similar frequencies of *POLG* variants, and while the role of this gene in PD remains under debate [38,39,40], our study validates the previous association. Both studies indicate that *POLG* is relevant as a PD gene, with odds ratios (ORs) and 95% confidence intervals consistently >1 and suggest that this association may be population-specific, with a predilection for Scandinavia. A potential caveat is the possible overrepresentation of PD subjects in our cohort and that of Anvret et al. [37], as both were drawn from the Stockholm area. However, several factors suggest distinct subject pools. Our cohort included individuals of mixed ethnicity (see Appendix A); however, ethnicity was not specified in the study by Anvret. Additionally, the mean ADx in our cohort was 62.95 ± 10.55 years, which coincided with the age of recruitment, ensuring a temporal alignment. In contrast, Anvret et al. [37] reported a mean age of diagnosis of 59.4 years and a mean collection age of 67.3 years, reflecting a disease duration of approximately 8 years at recruitment, compared to ~5 years in our cohort. These demographic and temporal differences, combined with the inclusion of subjects from diverse ethnic backgrounds in our study, further differentiates the two cohorts, despite their geographic proximity.

Beyond the observed genetic contributions, our study provides new insights into the *APOE*–*POLG* interaction, particularly its influence on age at diagnosis and CSF endophenotypes. E3 carriers with rare *POLG* variants show a protective effect, leading to a later diagnosis of PD. A striking discovery is the substantial increase in beta-amyloid levels in CSF among *APOE* E3 carriers with the most common *POLG* haplotype, compared to their E4 counterparts. Therefore, *APOE*–*POLG* interaction not only influences the age of diagnosis of PD but also impacts cerebrospinal fluid (CSF) biomarkers, offering potential avenues for early diagnosis and targeted therapies. *APOE* E4 has been modestly associated with lower cognitive scores, though not without discrepancies [38,39,40,41,42]. It has also been linked to rapid motor progression in PD [43], and genome-wide analyses have identified it as a determinant of mortality [44]. However, its interaction with *POLG* remains unexplored.

Our screening identified *ATXN2* as the second most frequent gene of interest, with allele sizes ranging from 27 to 37 CAG repeats, and the most frequent were those sized with 27 repeats. Alleles with 27 CAG repeats are generally considered low or no risk for ALS [45].

However, the risk for ALS diminishes as the CAG repeat length approaches the SCA2 threshold of 33 repeats, a range where the repeat length is conclusively pathogenic for the Central Nervous System, leading to SCA2 [46], and where genetic overlap with FTD has been identified [47]. While this relative risk estimate may suggest a potential pan-neuronal toxicity, the association appears to be ALS-specific and not necessarily relevant to PD. We presented a case of PD with homozygous *ATXN2* intermediate alleles (27/27 CAG repeats). A prior case by [48] reported homozygous 31/31 CAG repeats as non-pathogenic but conferring ALS risk, prompting to propose varied mechanisms by which these variants may contribute to *ATXN2*-spectrum diseases. The frequency of these co-occurrences in gnomAD v4.10 is 27/27 repeats = 4 × 10^−4^ vs. 3 × 10^−3^ in our population. The summation for each genotype is 54, corresponding to *ATXN2* risk genotypes, such as 22/32 CAG repeats. In this patient, homozygosity could exacerbate neurodegeneration via a gene dosage effect, a pseudo-recessive inheritance pattern where bi-allelic intermediate alleles amplify disease risk, or through RNA gain-of-function effects and disrupted stress granule dynamics. These mechanisms may synergize in homozygous states, driving severe motor and cognitive phenotypes. This case highlights the need to investigate diverse inheritance patterns and the molecular mechanisms by which intermediate alleles contribute to disease heterogeneity in *ATXN2*-related disorders.

Notably, inconsistencies have been reported regarding the association between *ATXN2* intermediate alleles and PD. Gispert et al. [49] analyzed a large cohort of ~1500 PD subjects and found a significant enrichment in 27–28 CAG alleles in the Düsseldorf subgroup. However, the overall risk of sporadic PD remained unchanged, as no such enrichment was observed in the Frankfurt or Tübingen subgroups from the same study. In contrast, Yamashita et al. [50] concluded that *ATXN2* polyQ expansion is a specific predisposing factor for PD, reporting that alleles with ≥24 repeats were significantly enriched in PD patients with typical L-DOPA-responsive phenotypes. However, Wang et al. [19] refuted this finding using a large dataset of >12K PD cases from the Genetic Epidemiology of PD Consortium (GEOPD) and applying the same cutoff (≥24 repeats). The unbalanced control/subject ratio (0.66) in the Wang et al. study may have limited the precision of their estimations. Nevertheless, their findings were ultimately taken as evidence to dismiss the routine screening of SCA2 in PD patients [51]. Ataxin-2 remains a conundrum, with emerging studies reinforcing its role in PD through diverse mechanisms. These include novel variants, full expansion, and recessive inheritance, such as the 9-bp duplication, all highlighted here; *ATXN2* double dosage [52]; and intermediate or fully expanded CAG repeats (35–39 CAG repeats), which have recently been linked to the parkinsonian spectrum [26,53,54].

*C9ORF72* is rarely, if ever, associated with PD [55,56,57,58,59,60], making the presence of six alleles displaying the characteristic saw-tooth-like pattern particularly striking. We assumed the alleles range from intermediate to moderately long to address the contentious dilemma between evidence and consensus, supported by the empirical observation that long tails were absent in the fragment analysis compared to the long alleles typically observed in ALS (Appendix A). Additionally, two other alleles, with repeat lengths of 30 and 35, completed the C9 cohort. Notably, 66% of expanded alleles shared the risk-associated (T) haplotype at rs3849942 and exhibited detectable methylation at the CpG island. All C9 carriers showed pure PD, and the clinical and imaging data were typical for PD with DAT striatal loss and also cerebellar degeneration. While previous studies in Sweden have excluded *C9ORF72* as a PD gene based on the saw-tooth pattern [61], further investigations using orthogonal techniques, such as Asuragen or Long-Read Sequencing, are necessary to accurately estimate the size and classification of these likely intermediate/moderately long, mutatis mutandis, alleles in our PD cohorts. Our cases challenge the binary all-or-nothing criteria by demonstrating the presence of a saw-tooth pattern.

Notably, one carrier of ≥60 C9RE repeats exhibited co-occurring digenic inheritance with an intermediate *TBP* allele, further underscoring the complexity and significance of this case series. Intermediate or incomplete penetrant *TBP*_41–46_ alleles have also been identified in cases of digenism associated with *STUB1* mutations. These findings are now recognized as defining two distinct spinocerebellar autosomal recessive type 16 (SCAR16 or SCA17-DI) entities separated from SCA48 [62].

In our study, although *TBP* was not statistically significant, it remains clinically relevant, as two patients were identified with full penetrant expansions. Precisely assigning risk helps determine the proportion of affected individuals, but even a single case with a pathogenic variant carries significant translational value for advancing precision medicine. While statistical significance aids in assigning risk within a population, it often overlooks the importance of low-prevalence variants. Non-significance is frequently misinterpreted as no effect; however, when a variant is known to be pathogenic, its clinical relevance is undeniable, regardless of statistical outcomes. This underscores the crucial need to integrate statistical analysis with clinical evidence to accurately assess its impact on population-level risk and individual patient care. In a clinical context, we treat patients, not *p*-values.

The association of intermediate expansions in *TBP* with PD has been a topic of significant debate. It is important to highlight that the inconsistency in defining thresholds for TBP intermediate and incomplete penetrance is not merely a matter of debate but often reflects a superficial approach to the subject. For instance, while one publication after another cites thresholds of 41–46 repeats as the range for intermediate penetrance [62], other studies arbitrarily extend this range to 49 repeats [63,64]. Such ad hoc adjustments not only lack rigorous justification but also complicate consistent data interpretation and cross-study comparisons. However, expert consensus established the ranges as 43–48 repeats for reduced penetrance and 49–66 repeats for full penetrance [65]. Two meta-analyses, conducted by the GEO-PD consortium [19] and Rossi et al. (MDSGene Task Force) [66], further addressed this question. The GEO-PD study used a threshold of 42–47 repeats (based on SCA17 criteria) and found no evidence of an association with PD. However, Rossi et al. [66], redefining the threshold, observed that pure parkinsonism was more prevalent in ATX-TBP patients with 41–45 repeats, while those with ≥46 repeats more frequently presented with a complex phenotype characterized by mixed movement disorders. An updated genotype–phenotype assessment for ATX-TBP is presented, proposing new repeat expansion cutoffs: reduced penetrance (41–45 repeats) and full penetrance (46–66 repeats). These revisions carry diagnostic and counseling implications and may inform future clinical trial protocols [66].

In our study, the mutation range spanned from reduced to full penetrance (41–48 CAG repeats), including one case with dual inheritance. The estimated sample size is promising for detecting small effects in our new cohorts, laying a strong foundation for impactful future studies.

The association of the *ApoE*-E4 haplotype with the presence of long *TOMM40* alleles has been well-documented, with previous studies, such as Roses et al. [36], highlighting this linkage and relevance in Alzheimer’s Disease. Our findings confirm this strong association with long and very long alleles being overrepresented in the *ApoE*-E4 haplotype. However, the association of specific haplotypes with PD is inconsistent in our and other studies [67,68,69]. Surprisingly, long genotypes (L-VL) were significantly associated with PD, which may initially be attributed to other effects, such as the enrichment of *APOE*-ε4 alleles and the effects of LD with E4. However, the S-L genotype, despite similar ε4 enrichment, showed no significant association. The most plausible explanation for this finding is the observed HWE disequilibrium in controls for the same genotype, which contributed to deviations. Notably, when the expected frequency is adjusted, this association disappears. Therefore, the deviation in the control group raises questions about whether the association is either artifact-driven (the result of comparing a biased control population to PD cases) or disease-driven (a reflection of a genuine biological predisposition in PD patients), which would suggest that the association is spurious and that *TOMM40* is irrelevant to PD. Beyond any speculation, these results should be interpreted with caution and require further confirmation in a more specifically designed control cohort.

Independent of genetic association, we found that *TOMM40* in our cohort significantly influences severity and long-term progression of PD, outperforming the more uneven influences of *APOE* and *POLG*. Its effects span nearly all UPDRS subdomains, making it a stronger predictor of disease progression and severity. In follow-up studies, short and long alleles have been shown to predict cognitive performance over time [69]; however, this was not supported by our observations but still validate *TOMM40*’s role in PD.

Gene expression studies, using cloned LD regions encompassing the *TOMM40*-*APOE-APOC* genomic signature, have revealed that enhancers within *TOMM40* regulate the promoter activity of both *TOMM40* and *APOE* in a haplotype- and cell-type-specific manner. This regulatory activity is further influenced by the length of the poly-T repeat, with enhanced expression notably observed in neurons [70].

Our analysis reveals a disequilibrium in some genotypes when using the unified long haplotype (L), contrasting with studies that divide it into subgroups (La and Lb) [71,72]. This division creates artificial haplotypes and genotypes, diluting potential signals of disequilibrium by generating lower expected frequencies that favor the Hardy–Weinberg Equilibrium (HWE). When calculating HWE without separating the haplotypes, we observed expected frequencies of zero for some genotypes, resulting in infinitum chi-square values and complicated interpretations. Maruszak et al. [71] state that *La* and *Lb* have similar effects. Unifying haplotypes better reflects the genetic structure, as demonstrated in Australian cohorts [68], where no HWE disequilibrium was seen (calculations were made by us). Despite methodological challenges, we recommend harmonized approaches in future studies to avoid artifacts and improve comparability.

Repeat length variations in *ATXN3*, *PRNP*, and *CACNA1A* were irrelevant for PD in our cohort.

Mutations in *ATXN3*, *PRNP*, and *CACNA1A* have been implicated in PD and parkinsonism through diverse mechanisms and clinical presentations in isolated cases. *ATXN3*, associated with Machado–Joseph disease (SCA3), has been reported in cases of atypical parkinsonism with L-dopa responsiveness, especially in patients with relatively low repeat numbers [21,72,73,74]. *CACNA1A*, linked to SCA6, has been described in cases of parkinsonism combined with cerebellar ataxia, with both L-dopa-responsive and non-responsive presentations, suggesting variability in dopaminergic dysfunction [75]. In contrast, polymorphisms in *PRNP*, including codon 129 variations, show no association with PD genetics across diverse populations [76].

We present observations that underscore the meaningful contribution of these genes rather than mere associations. Age at diagnosis was correlated with POLG, male sex, UPDRS, H & Y, MoCA, and tau levels. Among these, tau levels were validated through regression analysis, while repeat expansions exhibited a trend toward association. Additionally, ADx correlated with DD. While many of these correlations (sex, UPDRS, H & Y, MoCA, and tau) have been confirmed with age at onset in other cohorts [77,78,79], the positive correlation between *POLG* variants and age at diagnosis is particularly intriguing.

Our analysis reveals that carriers of more frequent *POLG* variants experience faster disease progression, as measured by the UPDRS, with significant differences across all UPDRS subdomains. This highlights a potential role for *POLG* in influencing motor symptom severity and progression in PD, warranting further investigation into its pathogenic contribution. Furthermore, *TOMM40* demonstrates superior predictive value over *APOE* and *POLG* when grouping PD individuals for progression analyses. Importantly, the *TOMM40* effect is not a simple surrogate for *APOE*-related mechanisms, as the PD profiles of individuals differ significantly, underscoring its distinct role in disease progression.

### Limitations

Our study was technically robust: all samples were tested in a single facility with rigorous quality control, internal standards consistently applied, and a control/patient design offering substantial predictive value. However, several limitations should be acknowledged. Increasing the sample size of the PD subjects, given the wide range of odds ratios, would enhance the power of our study and improve the precision of the associations. The low or null marginal frequencies of some variants may affect the robustness of these associations. Using ADx instead of age at onset may introduce inaccuracies in estimating the true onset of the disease. Likewise, missing values affect the stability of ANOVA, regression analyses, and correlations, posing a significant challenge. This is a common issue in tertiary centers and clinics managing large datasets, where incomplete data can undermine the robustness and reliability of statistical analyses. Addressing this limitation is critical to ensure accurate and meaningful interpretations of the findings. The lack of nuclear imaging analyses limits our ability to closely examine neurodegeneration and correlate it with genetic or clinical findings. Additionally, the absence of sequencing for some samples prevented us from confirming both repeat length and specific genetic variants or detecting triplet interruptions, which are known to modify disease risk and progression. For *C9ORF72*, while three-primer G4C2-Repeat Primed (RP)–PCR coupled to capillary electrophoresis is reliable for sizing alleles with ~24–45 repeats, it is less accurate for larger expansions, which could be better resolved using methods like Asuragen (Limit Of Detection ~145 rep) or long-read sequencing. However, we followed the routine practices commonly conducted in most research labs, given the prohibitive costs of both techniques (USD 50 and USD 3500 per individual, respectively) compared to USD 10 for conventional (RP)-PCR. Furthermore, our study was conducted at a time when these techniques were not yet available, and some of our DNA samples were of suboptimal quality. The confirmation of low-risk alleles (*ATXN2*: 27 repeats, *TBP*: 41–42 repeats, *C9ORF72*: 21–30 repeats) through functional studies is essential, along with observational analyses at the neuronal level, to establish their correlation or potential causal role. We also did not evaluate newly discovered non-repeat expansions, such as GGC repeats in *ZFHX3* [80], which may explain additional PD cases or interact with other mutations to influence phenotypes. The longitudinal analysis, with only two time points, may have failed to capture stable or nuanced changes over time, leaving the data vulnerable to experimenter or clinician biases. Moreover, the use of blood bank controls instead of a population-representative control group may have introduced a recruitment bias, as blood bank participants often include undiagnosed cases. Finally, while our study identified significant associations, these do not establish causality. Functional research is essential to validate these findings and explore the underlying mechanisms, especially for the genetic variants and repeat expansions highlighted in this study. However, while these limitations are acknowledged, they do not compromise our conclusions given the screening nature of our study, as our findings align with and are validated by previous studies that have established the role of these genes.

## 4. Materials and Methods

### Clinical Description of the Cohort

The initial clinical cohort included 192 participants, of whom 161 (83.85%) were diagnosed with PD. The remaining 31 (16.15%) had parkinsonian-related disorders, including GBA1-associated parkinsonism (10; 5.21%), corticobasal degeneration/syndrome (CBD/S; 6, 3.13%), fragile X-associated tremor/ataxia syndrome (FXTAS; 5, 2.60%), progressive supranuclear palsy (PSP; 5, 2.60%), multiple system atrophy—parkinsonian type (MSA-P; 2, 1.04%), multiple system atrophy—cerebellar type (MSA-C; 1, 0.52%), Niemann—Pick Disease with parkinsonism (NPD; 1, 0.52%), and PD with dementia (1, 0.52%). For this study, only PD subjects (N = 161) were retained (Figure 1a). Genetic aggregation in Parkinson’s disease (PD) in our cohort was determined based on Appendix A.

All 161 PD participants provided both oral and written informed consent. The study was approved by the Swedish Ethical Review Authority (Etikprövningsnämnden, Dnr 2016/19-31/2 and Dnr 2016/2503-31/2, approved on 21 March 2016). The diagnosis was established based on the UK PD Society Brain Bank (UK PDSBB) criteria.

A full description of the methods is presented in the Appendix A.

## 5. Conclusions

Our study highlights the significant contributions of *POLG*, *ATXN2*, and *C9ORF72* to PD in Sweden, with these genes being more frequent risk factors compared to other PD-related genes. The contribution of each gene can be broadly assessed as a population-level risk. Pathogenic variants in *ATXN2*, *C9ORF72*, *TBP*, and *POLG* are significantly associated with an increased risk of PD, with OR = 2.62 (95% CI: 1.80–3.81, *p* = 0.00001) (Appendix A). *TOMM40* genotypes, particularly those involving rare variants, are linked not only to the genetic architecture of PD but also to disease progression, with their role extending beyond the well-documented *APOE* E4 effect. While these genes are considered risk factors, their relevance to translational research is critical for stratifying PD cohorts to better understand disease presentation, long-term progression, and therapeutic implications. Other repeat expansion genes, including *ATXN3*, *CACNA1A*, and *PRNP*, typically associated with parkinsonian disorders, do not exhibit an association with PD in our cohort. Similarly, the contribution of *TBP* to PD appears negligible at the population level; however, it remains clinically significant, as a subset of patients carried anomalous repeat expansions that warrant further clinical investigation. For *ATXN2* and *C9ORF72*, where their roles in PD are not fully established, further studies are needed to explore the contribution of intermediate and long alleles as risk factors or potential monogenic determinants of the disease. The application of next-generation sequencing technologies will be vital for determining the size, sequence content, and pathogenic effects of *C9ORF72* intermediate alleles on PD-related structures. While traditional approaches focusing on DNA, mRNA, and CSF biomarkers remain central to therapeutics and translational research, emerging avenues, such as circRNA and cell-free DNA, offer significant opportunities for advancing biomarkers and therapeutic strategies in PD and precision medicine. Finally, our findings emphasize the distinctive genetic landscape of PD in Sweden, highlighting the value of population-specific studies to unravel genetic contributions and to refine diagnostic and therapeutic approaches.

## Figures and Tables

**Figure 1 ijms-26-09213-f001:**
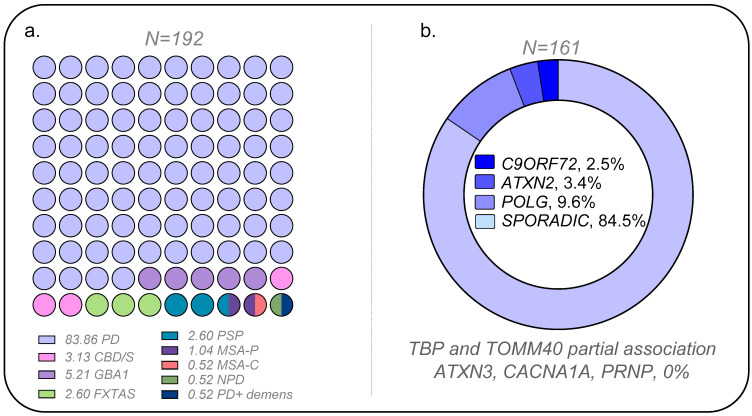
Distribution of genetic and clinical contributions to the cohort. (**a**) Clinical classification of the initial cohort (N = 192). Each circle represents the proportional percentage of each phenotype within the cohort. The majority of cases are classified as PD (PD, 83.86%, blue), followed by GBA-associated parkinsonism (5.21%), corticobasal degeneration/syndrome (CBD/S, 3.13%, pink), FXTAS (2.60%), progressive supranuclear palsy (PSP, 2.60%), multiple system atrophy—parkinsonian type (MSA-P, 1.04%), multiple system atrophy—cerebellar type (MSA-C, 0.52%), Niemann–Pick disease (NPD, 0.52%), and PD with dementia (0.52%). (**b**) Proportion of cases associated with different genetic contributors in the PD subset (N = 161). Sporadic cases account for the majority (84.5%), followed by *POLG* (9.6%), *ATXN2* intermediate expansions (3.4%), and *C9ORF72* (2.5%). TBP and *TOMM40* showed partial associations, while *ATXN3*, *CACNA1A*, and *PRNP* had no detected associations (0%). The data underscore the predominance of sporadic PD cases, while highlighting the contribution of specific genetic factors to the overall cohort composition.

**Figure 2 ijms-26-09213-f002:**
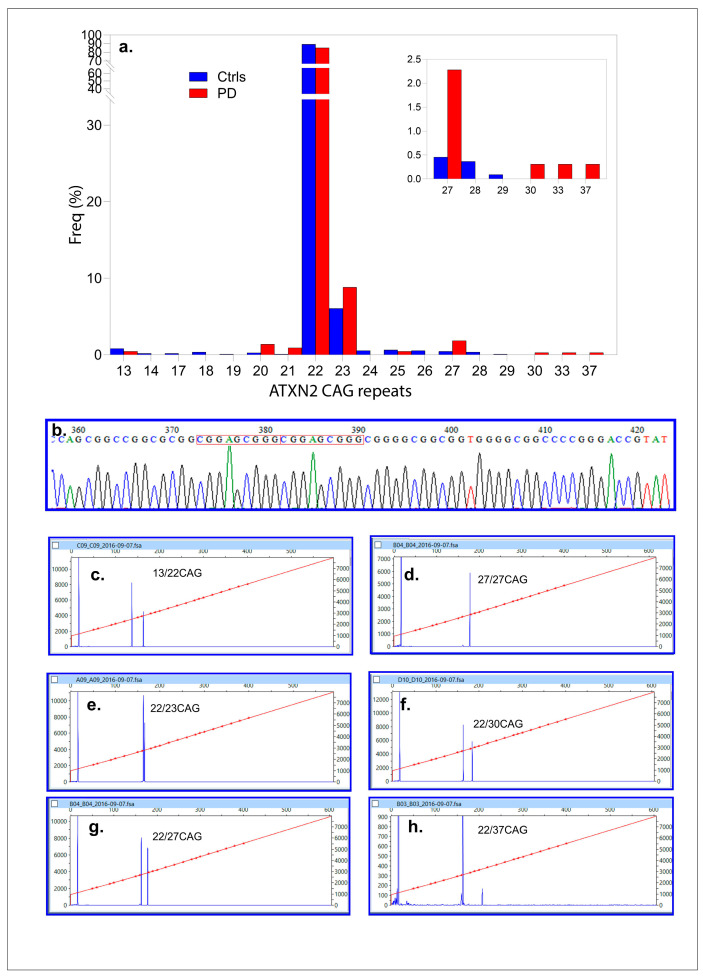
Analysis of *ATXN2* CAG repeat expansions in PD cases and controls. (**a**) Histogram showing the percentage of individuals with specific *ATXN2* CAG repeat lengths in PD cases (red bars) and controls (blue bars). The main panel highlights the distribution across the full range of CAG repeats, with 22 repeats being the most frequent in both groups. The inset focuses on alleles with longer repeat lengths (≥27 repeats), where an enrichment of ≥27 repeats of the intermediate allele *ATXN2* is observed in PD cases compared to controls. (**b**) Sanger sequencing chromatogram of g271_279delinsCGGAGCGGG duplication. The boxed area highlights the duplicated repeat sequence for the wild type and for PD with the 9-bp duplication. (**c**) Electropherograms obtained from capillary electrophoresis illustrating repeat size determination for representative samples. Each panel shows the genotypes for CAG repeat expansions. Peaks correspond to the fragment sizes of PCR products. (**c**) displays a genotype with 13 and 22 CAG repeats, (**d**) represents a homozygous case with 27 CAG repeats, (**e**) shows 22 and 23 repeats, (**f**) illustrates 22 and 30 repeats, (**g**) corresponds to 22 and 27 repeats, and (**h**) indicates 22 and 37 repeats. The red line within the fragment analysis traces represents the size calibration curve.

**Figure 3 ijms-26-09213-f003:**
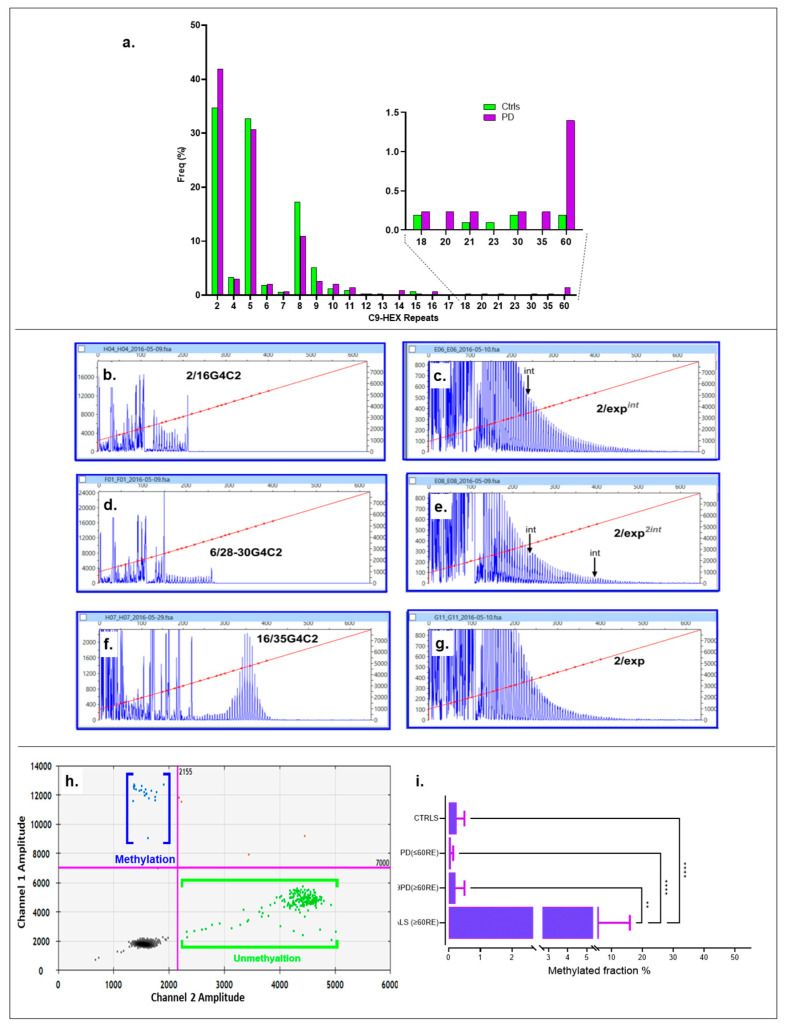
Analysis of *C9ORF72* repeat expansions, methylation status, and associated patterns in controls and PD subjects. (**a**) Frequency distribution of *C9ORF72* repeat lengths in control (green) and PD (purple) groups. The inset shows the proportion of cases and controls with expanded alleles (18–60 G4C2 repeats). (**b**–**g**) Capillary electrophoresis traces showing representative examples of *C9ORF72* alleles: (**b**) Normal allele with 2 repeats and 16 repeats (2/16G4C2). (**c**) Normal allele with 2 repeats and low-range expanded allele (2/exp) and one interruption indicated by the arrow (int). (**d**) Heterozygous allele with both normal (6/28-30G4C2) and expanded repeats sized with 30 repeats. (**e**) Normal allele with 2 repeats and moderately expanded allele with repeats showing interrupted sequences (int). (**f**) Normal allele with 16/35G4C2 repeats. (**g**) Normal allele with 2 repeats and low-range expanded allele (2/exp). (**h**) Two-dimensional ddPCR scatter plot showing methylation analysis in PD *C9ORF72* repeats. Blue dots represent methylated alleles, while green dots indicate unmethylated alleles. The clustering along the axes reflects the degree of methylation. (**i**) Bar plot summarizing the percentage of methylated *C9ORF72* alleles in controls and PD samples. Statistically significant differences between groups are highlighted (** *p* = 0.01, **** *p* = 0.0001). The red line within the fragment analysis traces represents the size calibration curve.

**Figure 4 ijms-26-09213-f004:**
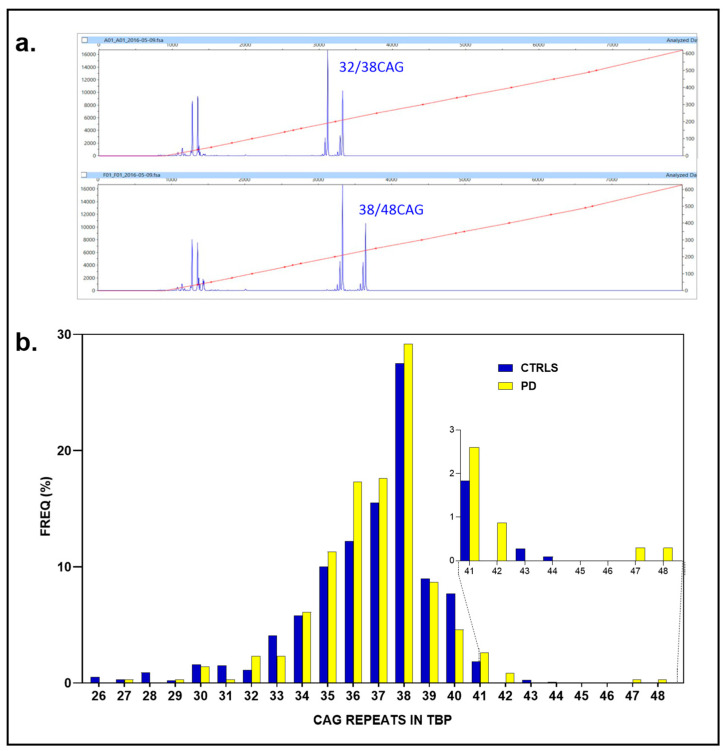
Analysis of CAG repeat length in *TBP* gene for PD. (**a**) Representative electropherograms showing CAG repeat expansions in the *TBP* gene. The top panel displays a case with 32/38 CAG repeats, while the bottom panel shows a case with 38/48 CAG repeats. (**b**) Distribution of CAG repeat frequencies (%) in PD cases (yellow) and controls (blue). The main histogram depicts the modal distribution centered around 38 repeats, with longer repeats (≥41 CAG repeats) displayed in the inset graph, emphasizing their relative enrichment in PD cases compared to controls. The red line within the fragment analysis traces represents the size calibration curve.

**Figure 5 ijms-26-09213-f005:**
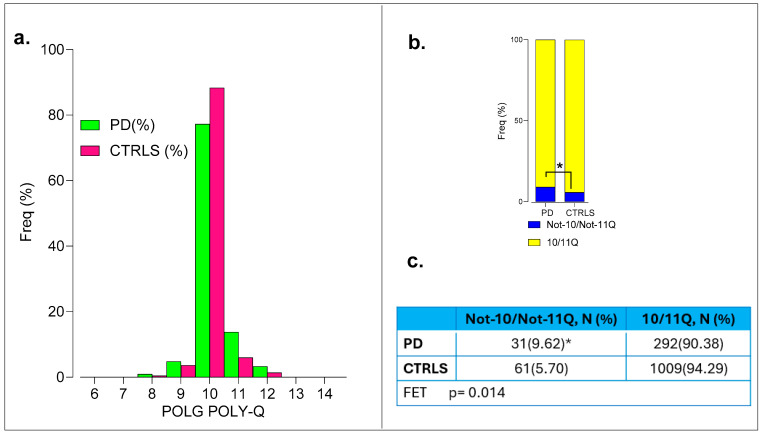
Association of *POLG* rare alleles with Parkinson’s disease. (**a**) Frequency distribution of *POLG* Poly-Q repeat lengths across PD cases (green) and control individuals (pink). Most alleles fall within the 10Q category for both groups. A slight shift is observed in the distribution of the 11Q alleles. (**b**) Proportions of 10/11Q carriers (yellow) and Not-10/Not-11Q carriers (blue) among PD and CTRL cohorts. The PD group exhibits a significant enrichment in the Not-10/Not-11Q genotype compared to the controls (*p* = 0.014), as indicated by the Fisher’s exact test. (**c**) Summary table of genotype frequencies for Not-10/Not-11Q and 10/11Q categories in PD and CTRL cohorts. A statistically significant association is observed for the Not-10/Not-11Q genotype in the PD cases (9.62%) versus controls (5.60%) (*p* = 0.014, * *p* = 0.05). Frequencies in the table are presented as percentages (%), but the analysis was conducted with counts to avoid inflated estimations.

**Figure 6 ijms-26-09213-f006:**
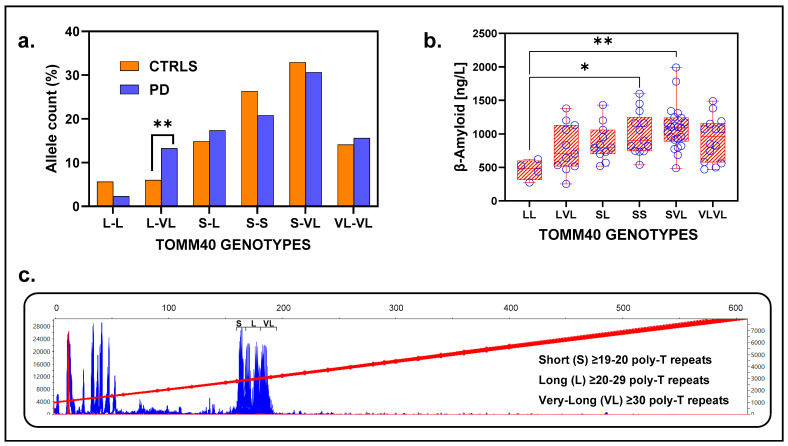
*TOMM40* genotypes and their association with allele distribution and β-amyloid levels. (**a**) Allele distribution across *TOMM40* genotypes. Bar plot showing the allele frequency distribution of *TOMM40* genotypes between PD cases (blue) and controls (orange). A significant difference (*p* = 0.01) is observed for the L-VL genotype, with a higher frequency in PD cases compared to controls. (**b**) β-amyloid levels by *TOMM40* genotypes. Box plot illustrating β-amyloid levels across *TOMM40* genotypes. Significant differences are noted, with elevated β-amyloid levels in S-VL (** *p* = 0.01) and S-S (* *p* = 0.05) genotypes compared to L-L genotypes. (**c**) Electropherogram of TOMM40 Poly-T repeat regions. Overlayed electropherogram of 600 samples showing the distribution of poly-T repeat lengths corresponding to *TOMM40* genotypes. The ranges for short (S, 19–20 repeats), long (L, 20–29 repeats), and very long (VL, ≥30 repeats) are indicated. The red line within the fragment analysis traces represents the size calibration curve.

**Figure 7 ijms-26-09213-f007:**
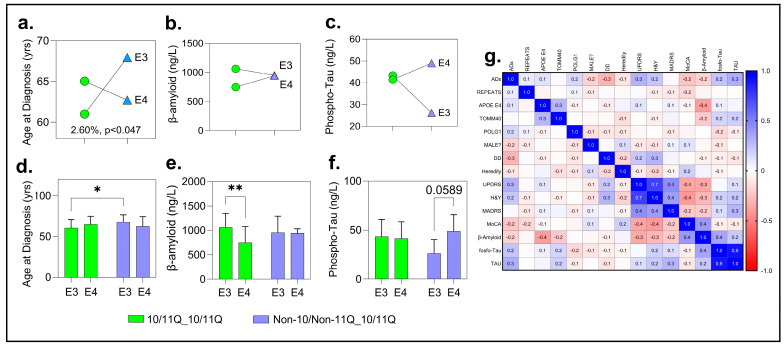
Interaction between APOE status, POLG genotypes, biomarkers, and clinical measures with PD based on a 2 × 2 ANOVA. (**a–c**) Interaction plots derived from a 2 × 2 ANOVA analysis showing the effects of APOE genotypes (E3 vs. E4) and POLG genotypes (10/11Q_10/11Q, Non-10/Non-11Q_10/11Q) on (**a**) age at diagnosis (years), (**b**) β-amyloid levels (ng/L), and (**c**) phospho-tau levels (ng/L). A significant interaction effect is noted in (**a**), where the percentage and *p*-value indicate explained variance and statistical significance. (**d–f**) Bar plots comparing biomarkers and clinical measures between APOE E3 and E4 carriers, further stratified by POLG genotype. Green bars: carriers with the 10/11Q_10/11Q genotype; Blue bars: carriers with the Non-10/Non-11Q_10/11Q genotype. Error bars denote standard deviation (SD). (**d**) Age at diagnosis: significant effect between APOE E3/E4 and POLG genotypes (* *p* = 0.05). (**e**) β-amyloid levels: significant main effect of APOE E4 (** *p* = 0.01). (**f**) Phospho-tau levels: trend-level difference (*p* = 0.0589). (**g**) Correlation matrix illustrating relationships between key clinical, genetic, and biomarker variables in the cohort. Color coding indicates correlation coefficients (blue = positive correlation; red = negative correlation), with stronger correlations represented by more intense colors. Variable labels: ADx, age at PD diagnosis; REPEATS, repeat size; APOE E4, presence of the E4 allele; TOMM40 and POLG, genetic markers; MALE, male sex; DD, disease duration; Heredity, family history of disease; UPDRS, Unified Parkinson’s Disease Rating Scale; H&Y, Hoehn and Yahr stage; MADRS, Montgomery–Åsberg Depression Rating Scale; MoCA, Montreal Cognitive Assessment; β-amyloid, Phospho-Tau, and Tau: biomarker levels.

**Figure 8 ijms-26-09213-f008:**
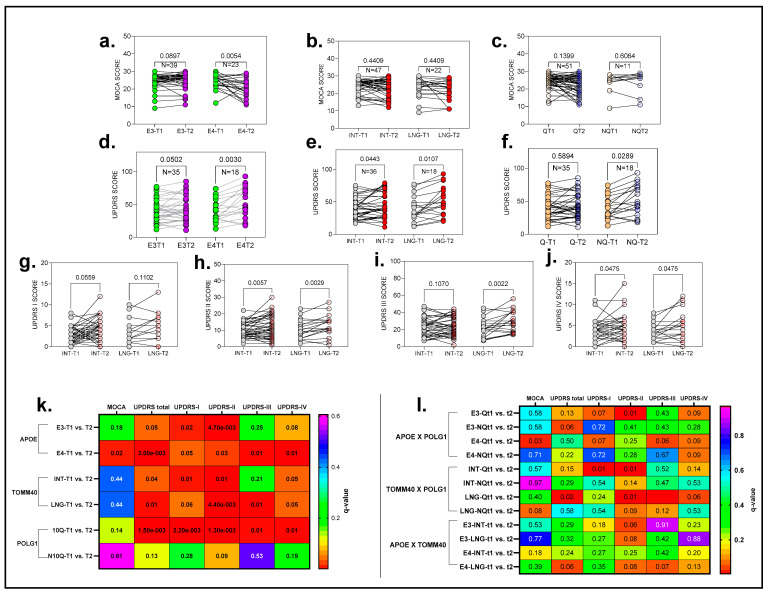
Results from a comparative longitudinal analysis of haplotype variations within the genes *APOE*, *TOMM40*, and *POLG* and their associations with clinical parameters. The subjects were evaluated for changes in haplotype groups related to each gene, along with clinical assessments using MoCA and MDS-UPDRS. The top graphs (**a**–**j**) display before-and-after plots representing haplotype-associated clinical parameters at the two time points, illustrating changes between them. Panels (**k**) and (**l**) show heat maps summarizing the haplotype interactions with the clinical measures. The rows in the heat maps correspond to haplotypes within *APOE*, *TOMM40*, and *POLG* at both T1 and T2, while the columns represent clinical measures (MoCA, total UPDRS, and individual UPDRS items). The color scale represents q-values derived from false discovery rate (FDR) corrections. Warmer colors (red, orange, and yellow) indicate lower q-values, reflecting more significant interactions. Cooler colors (green, blue, and magenta) correspond to higher q-values, indicating less significant findings. The q-values are mapped directly to the scale on the right, ranging from 0.0 to 0.6. This analysis provides insights into the correlation between genetic haplotypes and clinical progression over time within the studied cohort.

## Data Availability

Any data related to this article that are not published within it may be requested through collaboration. Additional access may be granted at the discretion of the investigators, with a clear intent for its use. Otherwise, all data are available on the KI servers.

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
