# Peer review of "Repeat Variants, Biomarkers, and Molecular Signatures in Parkinson’s Disease: ATXN2, ATXN3, CACNA1A, PRNP, TBP, C9ORF72, TOMM40, APOE, and POLG—A Swedish Perspective"

_ijms, 2025, doi:10.3390/ijms26189213_

Round 1

Reviewer 1 Report

Comments and Suggestions for Authors

This study examines how several genetic factors, including nucleotide repeat expansions, may contribute to Parkinson’s disease (PD) in a Swedish cohort. The authors provide a large dataset and interesting findings on the association of ATXN2, C9ORF72, and POLG1 with PD, as well as new gene interactions. The work could add valuable insight into the genetics of PD. However, before publication, key issues with methods and data reporting need to be addressed, along with minor errors that affect clarity and professionalism.

Major issues:

  1. The manuscript shows inconsistent reporting of p-values. Some are reported as exact values (e.g., P=0.0027), while others are given as thresholds (e.g., p≤0.05). For instance, in the association analysis of the POLG1 gene, the Abstract reports p≤0.05, Section 2.5 of the main text reports p=0.014 and p=0.01, and the legend for Figure 5 states p≤0.014. The authors should review the entire manuscript to ensure consistency in p-value reporting.
  2. In Section 2.1.1, the authors state that the methylation analysis of ATXN2 revealed it was “fully unmethylated.” However, in Section 2.1.4, they report that a patient with Progressive Supranuclear Palsy (PSP) “exhibited low but detectable methylation levels for ATXN2-AS (1%).” While these may refer to different molecular targets (ATXN2 sense strand vs. ATXN2-AS), the absolute term “fully unmethylated” in Section 2.1.1 is misleading. I suggest replacing it with the more accurate phrase “generally unmethylated.”

  1. The reported link between the TOMM40 L-VL genotype and Parkinson’s disease is seriously weakened by a major flaw: the control group fails Hardy–Weinberg Equilibrium (HWE) (p < 0.0001), as noted in Section 2.6. This suggests the controls are not a reliable baseline and the result may be spurious. Since the authors themselves question whether this is “artifact-driven,” the current conclusion about TOMM40 is unsupported. The authors should either re-analyze the data with a valid control group meeting HWE or clearly present this as a preliminary result needing independent confirmation.

  1. In the third paragraph of the Introduction, the sentence "...highlighting PD as a significant public health issue [7] reported prevalence rates between 115 and 136 per 100,000 individuals, confirming the disease's impact [8, 9]" is grammatically incorrect. The phrase "reported prevalence rates..." creates a run-on or fragment because it lacks a proper subject or connector after the citation "[7]." I suggest revising it to: “...highlighting PD as a significant public health issue, with reported prevalence rates between 115 and 136 per 100,000 individuals confirming the disease's impact.”

Minor issues:

  1. In the seventh paragraph of the Introduction, the phrase "...adding complexity to understanding their role in neurodegeneration (Wang et al., 2015)." Uses the author-year citation style, which is inconsistent with the numbered citation style used elsewhere in the manuscript.
  2. In the seventh and ninth paragraphs of the Introduction, “POLG1A” should be corrected to “POLG1.”
  3. In the ninth paragraph of the Introduction, the gene “c9orf72” should be corrected to “C9ORF72” to maintain consistent formatting.
  4. In the first sentence of the Results, “Figure1” should be corrected to “Figure 1.”
  5. In the second sentence of the Results, “The positive association for CAG repeats in TBP the associations were marginal” contains a grammatical error. It should be revised to “The positive association for CAG repeats in TBP was marginal.”
  6. In the first paragraph of the Results, “data no shown” should be revised to “data not shown.”
  7. In the first paragraph of the Section 2.1, “…controls DNAs were successfully genotyped” should be revised to “…control DNAs were successfully genotyped.”
  8. In the first paragraph of the Section 2.9, "MALE? " should be corrected to “MALE”. The question mark should be removed.

Author Response

Thank you very much for taking the time to review this manuscript. Please find the detailed responses below and the corresponding revisions/corrections highlighted/in track changes in the re-submitted files. All reviewer suggestions were proper and responded as they suggested.

This study examines how several genetic factors, including nucleotide repeat expansions, may contribute to Parkinson’s disease (PD) in a Swedish cohort. The authors provide a large dataset and interesting findings on the association of ATXN2, C9ORF72, and POLG1 with PD, as well as new gene interactions. The work could add valuable insight into the genetics of PD. However, before publication, key issues with methods and data reporting need to be addressed, along with minor errors that affect clarity and professionalism.

Major issues:

  1. The manuscript shows inconsistent reporting of p-values. Some are reported as exact values (e.g., P=0.0027), while others are given as thresholds (e.g., p≤0.05). For instance, in the association analysis of the POLG1 gene, the Abstract reports p≤0.05, Section 2.5 of the main text reports p=0.014 and p=0.01, and the legend for Figure 5 states p≤0.014. The authors should review the entire manuscript to ensure consistency in p-value reporting. Thank you for pointing this out. We agree with this comment. Therefore, we have corrected, except for those where the threshold has to be set and specified.
  2. In Section 2.1.1, the authors state that the methylation analysis of ATXN2 revealed it was “fully unmethylated.” However, in Section 2.1.4, they report that a patient with Progressive Supranuclear Palsy (PSP) “exhibited low but detectable methylation levels for ATXN2-AS (1%).” While these may refer to different molecular targets (ATXN2 sense strand vs. ATXN2-AS), the absolute term “fully unmethylated” in Section 2.1.1 is misleading. I suggest replacing it with the more accurate phrase “generally unmethylated.” Thank you for pointing this out. We agree with this comment. Therefore, we have corrected.

  1. The reported link between the TOMM40 L-VL genotype and Parkinson’s disease is seriously weakened by a major flaw: the control group fails Hardy–Weinberg Equilibrium (HWE) (p < 0.0001), as noted in Section 2.6. This suggests the controls are not a reliable baseline and the result may be spurious. Since the authors themselves question whether this is “artifact-driven,” the current conclusion about TOMM40 is unsupported. The authors should either re-analyze the data with a valid control group meeting HWE or clearly present this as a preliminary result needing independent confirmation. Thank you for pointing this out. We agree with this comment. Therefore, we have corrected.

  1. In the third paragraph of the Introduction, the sentence "...highlighting PD as a significant public health issue [7] reported prevalence rates between 115 and 136 per 100,000 individuals, confirming the disease's impact [8, 9]" is grammatically incorrect. The phrase "reported prevalence rates..." creates a run-on or fragment because it lacks a proper subject or connector after the citation "[7]." I suggest revising it to: “...highlighting PD as a significant public health issue, with reported prevalence rates between 115 and 136 per 100,000 individuals confirming the disease's impact.” Thank you for pointing this out. We agree with this comment. Therefore, we have corrected.

Minor issues:

  1. In the seventh paragraph of the Introduction, the phrase "...adding complexity to understanding their role in neurodegeneration (Wang et al., 2015)." Uses the author-year citation style, which is inconsistent with the numbered citation style used elsewhere in the manuscript. Thank you for pointing this out. We agree with this comment. Therefore, we have corrected.  
  2. In the seventh and ninth paragraphs of the Introduction, “POLG1A” should be corrected to “POLG1.” Thank you for pointing this out. We agree with this comment. Therefore, we have replaced to HGNC Approved Gene Symbol: POLG.
  3. In the ninth paragraph of the Introduction, the gene “c9orf72” should be corrected to “C9ORF72” to maintain consistent formatting. Thank you for pointing this out. We agree with this comment. Therefore, we have corrected.
  4. In the first sentence of the Results, “Figure1” should be corrected to “Figure 1.” We have corrected.
  5. In the second sentence of the Results, “The positive association for CAG repeats in TBP the associations were marginal” contains a grammatical error. It should be revised to “The positive association for CAG repeats in TBP was marginal.” We have corrected
  6. In the first paragraph of the Results, “data no shown” should be revised to “data not shown.” We have corrected
  7. In the first paragraph of the Section 2.1, “…controls DNAs were successfully genotyped” should be revised to “…control DNAs were successfully genotyped.” We have corrected
  8. In the first paragraph of the Section 2.9, "MALE? " should be corrected to “MALE”. The question mark should be removed. We have corrected

Reviewer 2 Report

Comments and Suggestions for Authors

Laffita-Mesa and co-authors have extensively characterized the genomic landscape of repeat variants in genes crucially involved in neurodegenerative disorders. The paper is well written and careful analysis has been performed. 

I do not have major comments content-wise. I would recommend the authors an intense revision of the figures:

- Figure S1: replace homocigotes/heterocigotes to homozygotes/heterozygotes. 
- Very often the authors make extensive use of y-axis breaks which are not really useful nor have the point to be there.  (see figure 5a and 5b where the breaks are completely not-necessary)
- Same for the insets in the plots. 
- Figure 2: title on the x-axis is recommended. Facilitates the reader to immediately catch the meaning of the plot.  
- In Figure 3 the "int" uppercase is not explained in the figure legend. 

Other minor adjustments:

- page 4, section 2.1 ATXN2. The sentence: "The most frequent allele was 22 for both populations" it is not clear. Please rephrase so that is clear that the 22 indicates the length of the repeat.
- page 12, section 2.6: bold text is present in the paragraphs. Please revise this and other minor typos/formatting errors in the whole text. 

Author Response

Thank you very much for taking the time to review this manuscript. Please find the detailed responses below and the corresponding revisions/corrections highlighted/in track changes in the re-submitted files. All reviewer suggestions were proper and responded as they suggested.

Laffita-Mesa and co-authors have extensively characterized the genomic landscape of repeat variants in genes crucially involved in neurodegenerative disorders. The paper is well written and careful analysis has been performed. 

I do not have major comments content-wise. I would recommend the authors an intense revision of the figures:

- Figure S1: replace homocigotes/heterocigotes to homozygotes/heterozygotes. Thank you for pointing this out. We agree with this comment. Therefore, we have corrected.
- Very often the authors make extensive use of y-axis breaks which are not really useful nor have the point to be there.  (see figure 5a and 5b where the breaks are completely not-necessary)
- Same for the insets in the plots. Thank you for pointing this out. We agree with this comment. Therefore, we have corrected.
- Figure 2: title on the x-axis is recommended. Facilitates the reader to immediately catch the meaning of the plot.  Thank you for pointing this out. We agree with this comment. Therefore, we have corrected.
- In Figure 3 the "int" uppercase is not explained in the figure legend. Thank you for pointing this out. We agree with this comment. Therefore, we have corrected.

Other minor adjustments:

- page 4, section 2.1 ATXN2. The sentence: "The most frequent allele was 22 for both populations" it is not clear. Please rephrase so that is clear that the 22 indicates the length of the repeat. Thank you for pointing this out. We agree with this comment. Therefore, we have corrected.
- page 12, section 2.6: bold text is present in the paragraphs. Please revise this and other minor typos/formatting errors in the whole text. Thank you for pointing this out. We agree with this comment. Therefore, we have corrected. All the text was thoroughly improved and edited.

Round 2

Reviewer 1 Report

Comments and Suggestions for Authors

The revised manuscript has addressed most of the issues from the first round reviews, and is substantially improved. However, several issues need to clarify. Therefore, my recommendation is “accept after minor revisions”.

Several issues:

  1. In the first sentence of the Results, “Figure1” should be corrected to “Figure 1.”
  2. The manuscript shows inconsistent reporting of p-values. Some are reported as exact values (e.g., P=0.0027), while others are given as thresholds (e.g., p≤0.05). For example, in the association analysis of the POLG1 gene, the Abstract reports p≤0.05, Section 2.5 of the main text reports p=0.014 and p=0.01, and the legend for Figure 5 states p≤0.014. The authors should review the entire manuscript to ensure consistency in p-value reporting.

Author Response

We thank the reviewer for this important observation. Figure1. was corrected to Figure 1. The english language was edited and corrected by native speaker. All instances of p-values throughout the manuscript (including the Abstract, main text, tables, and figure legends) have been carefully reviewed and corrected to ensure consistency. Threshold expressions (e.g., p < 0.05, p ≤ 0.05) have been replaced with their exact values (p = 0.05...) as provided by the statistical software. This has now been standardized across the entire manuscript.